# pH-Sensitive Biomaterials for Drug Delivery

**DOI:** 10.3390/molecules25235649

**Published:** 2020-11-30

**Authors:** Shijie Zhuo, Feng Zhang, Junyu Yu, Xican Zhang, Guangbao Yang, Xiaowen Liu

**Affiliations:** 1Clinical Translational Center for Targeted Drug, Department of Pharmacology, School of Medicine, Jinan University, Guangzhou 510632, China; caracheuk@stu2018.jnu.edu.cn (S.Z.); zf1411947710@stu2018.jnu.edu.cn (F.Z.); yujunyu@stu2018.jnu.edu.cn (J.Y.); 2Collaborative Innovation Center of Radiation Medicine of Jiangsu Higher Education Institutions, State Key Laboratory of Radiation Medicine and Protection, School of Radiation Medicine and Protection & School for Radiological and Interdisciplinary Sciences (RAD-X), Soochow University, Suzhou 215123, China; yangguangbao@nwpu.edu.cn

**Keywords:** pH sensitive biomaterials, pH sensitive linkage bonds, drug delivery system

## Abstract

The development of precise and personalized medicine requires novel formulation strategies to deliver the therapeutic payloads to the pathological tissues, producing enhanced therapeutic outcome and reduced side effects. As many diseased tissues are feathered with acidic characteristics microenvironment, pH-sensitive biomaterials for drug delivery present great promise for the purpose, which could protect the therapeutic payloads from metabolism and degradation during in vivo circulation and exhibit responsive release of the therapeutics triggered by the acidic pathological tissues, especially for cancer treatment. In the past decades, many methodologies, such as acidic cleavage linkage, have been applied for fabrication of pH-responsive materials for both in vitro and in vivo applications. In this review, we will summarize some pH-sensitive drug delivery system for medical application, mainly focusing on the pH-sensitive linkage bonds and pH-sensitive biomaterials.

## 1. Introduction

Biomaterials have been widely developed as the carriers of various drugs for medical application [1,2,3,4]—of which pH-sensitive biomaterials represent a promising species, which can perform deformation or degradation after exposure to external acidic or alkaline environments. The pH-sensitive characteristics enable the changes in intramolecular or intermolecular forces of the formulations in external pH conditions, resulting in a trigger release of payloads [5,6,7]. There are basically two major different mechanisms to release the loading drugs in pH-sensitive drug delivery systems [8]: one is releasing the medicines generated from the changed hydrophobicity or a charge of carrier molecules by protonation/deprotonation induced by the external pH variation. The other is releasing the loading drugs based on pH-sensitive dynamic chemical bond cleavage [8].

Cancer is one of the leading causes of human death around the world [9,10]. In the past decades, nanomaterials based on the tumor microenvironment (TME) has been proposed to be a promising approach to improve the efficiency of cancer therapy [11,12]. TME constructs the external environment of tumor cells, which is closely related to the occurrence and metastasis of tumors [13,14]. The tumor microenvironment is composed of tumor cells, stromal cells, cytokines, and matrix. Due to the rapid growth and high volume expansion of tumor tissues, the tumor microenvironment is characterized by low oxygen [15,16,17] and mild acidity pH (6.5–6.9) [18]. With the better understanding of tumor cell biology, the tumor microenvironment is of great significance for the formation, development, metastasis, and other processes of tumors [19,20]. These abnormal microenvironments in return also provide the unique parameters for the designing of intelligent biomaterials for cancer therapy [21,22]. For control release of the therapeutics, many pH-responsive formulations have been designed based on the tumor acidic environment [23,24].

pH-sensitive biomaterials have been developed rapidly in the past decades and have made great achievements. As shown in Figure 1, we have summarized the number of publications about pH-sensitive biomaterials in the past 20 years (Figure 1), which increases every year. In this review, we focus on the pH-responsive linkage bonds and pH-sensitive nanomaterials for medical applications, especially for cancer therapy. Firstly, we summarize some common pH-sensitive biomaterials based on different types of pH-sensitive bonds. In the following, we summarize pH-sensitive nanomaterials for drug delivery purposes.

## 2. pH-Sensitive Bonds

The functionalization of biomaterials, which are covalently bonded to drug molecules, is the most common way to obtain intelligent biomaterials [25,26,27,28,29]. This type of carrier has high stability and can effectively prevent the premature release of drugs under physiological conditions [30]. However, in order to deliver drugs effectively, responsive biomaterials are showing great prospect [31,32,33]—of which, pH-responsive biomaterials equipped with acidic trigger release capacity have been widely used for the purpose. The pH-responsive biomaterials are expected to store the drugs in normal pH and release the drugs in a specific pH condition. To achieve this, pH-sensitive chemical bonds can be anchored within the drug carriers, and the configuration of the carriers can be deformation through the cleavage of the dynamic chemical bonds, so as to control release the payloaded drugs [8]. In this section, we will introduce the commonly used pH-sensitive chemical bonds (Table 1).

### 2.1. Imine Bonds

Imines are formed by condensation of primary amine with aldehydes or ketones. Imine bond can be hydrolyzed under weak acid conditions (pH ~6.8, near the pH of solid tumors), but its stability in physiological pH environment can be improved [34,35,36,37,38,39].

Yang et al. synthesized a polymer poly(ethylene glycol)-cholic acid grafted poly-L-lysine (PEG-PLL-CA) anchored with benzoimide, and demonstrated that polycationic micelles formed by self-assembly of PEG-PLL-CA could function as the carriers of therapeutics for tumor therapy with pH-responsive characteristics [40]. The imine bond is liable to hydrolyze under acidic conditions which can also be used to modify the surface of the nanocarriers [41], and the acid-sensitive release of the drugs on the carrier can be realized triggered in the acidic environment. Xu et al. synthesized the pentaerythritol tetra(3-mercaptopropionate)-allylurea-poly(ethylene glycol) (PETMP-AU-PEG) anchored with imide bonds. They demonstrated that PETMP-AU-PEG loaded with DOX were highly stable in the neutral environment. While under the environment of weak acid, the formulation can be a responsive release of DOX for antitumor outcomes by cleavage of imide bond [42].

### 2.2. Hydrazone Bonds

Hydrazone is formed by condensation of aldehydes, ketones, or hydrazine. Similar to imine bonds, the hydrazone bonds are also a class of covalent bonds which are acidic responsive [43,44,45,46,47]. As hydrazone bonds are easily hydrolytic under acidic conditions, they have been widely utilized in the construction of acidic sensitive carriers for responsive formulations [48,49].

Rihova et al. applied N-(2-hydroxypropyl) methylacrylamide (HPMA) as the backbone to prepare HPMA-DOX formulations anchored with hydrazine bonds, and they reported that the formulations could increase the in vivo circulation time of DOX for optimal targeting accumulate in the tumor and afterwards responsive release triggered by the acidic tumor environment [50]. Song et al. first synthesized the polymer PEG-DiHyd-PLA containing the hydrazone bond by open-loop polymerization, and then the polymeric micelles were formed by self-assembly for loading DOX. They demonstrated that the formulations could release DOX rapidly and completely under the specific acidic conditions of tumor tissues [51].

### 2.3. Oxime Bonds

Oxime is formed by condensation of aldehydes, ketones, or hydroxylamine. Oxime can hydrolyze into aldehydes, ketones, and hydroxylamines under acidic conditions [52,53]. A potential advantage of using oxime bond for tumor-targeted nanocarriers is the tunability of its acid lability by facile variation of the substituents [54].

Zhu and colleagues described a polymeric drug carrier system by self-assembly of the triblock copolymer (PEG-OPCL-PEG) which were synthesized by conjugating of hydrophilic poly(ethylene glycol) (PEG) and hydrophobic oxime-tethered polycaprolactone (OPCL), which was successfully prepared by reacting OPCL ligating with aldehyde-terminated PEG (PEG-CHO). During the self-assembly process, DOX as a drug model was loaded within the PEG-OPCL-PEG micelles. They demonstrated that the release rate of DOX in tumor tissue was significantly faster than in normal tissue, as a result of the acidic responsive of anchored oxime linkages within the micellers [53]. In another work, Zhu and colleagues described a flower micelle formulated by self-assembly of backbone-cleavable triblock copolymer polycaprolactone-oxime-poly(ethylene glycol)-oxime-polycaprolactone (PCL-OPEG-PCL) in aqueous solution, during which the DOX were encapsulated within the micelles in high efficiency. In addition, the DOX loaded micelles had good stability in a neutral environment in vitro studies, but, under the condition of weak acid, DOX could be quickly released for the pH responsive characteristics and thus perform excellent antitumor outcomes [55].

### 2.4. Amide Bonds

Amide bond is formed by the substitution of the hydroxyl group in the carboxylic acid group by amino groups or hydrocarbon amino groups [53,56,57]. Amide bond can also be cleavage under acidic conditions [58,59,60,61,62], which are most widely used for designing pH-responsive formulations.

Gu et al. synthesized pH-sensitive polymeric micelle (mPEG-pH-PCL) through self-assembly of methoxy poly(ethylene glycol)-b-poly(ϵ-caprolactone) copolymer anchored with citraconic amide as a pH-sensitive bond, and DOX were efficiently encapsulated in the micelle during assembly. They demonstrated that the DOX released from mPEG-pH-PCL micelles in pH 5.5 solution was much faster than that of in pH 7.4, indicating a beneficial characteristic for drug targeting release in acidic tumors [63]. Forrest et al. used amide bonds to connect hyaluronic acid (HA) with DOX to form the HA-DOX conjugate. In the blood neutral environment, HA-DOX has good stability and biocompatibility during the circulation process, thus reducing dose-limiting cardiac toxicity and minimal toxicity observed in other normal tissues. However, once the therapeutic conjugation accumulated in the acidic environment of tumor tissue, the acidic hydrolysis of amide bond enabled DOX specific release, thus achieving better therapeutic effects [64].

### 2.5. Acetals

Acetals (including ketals) are promising acid-sensitive linkages for responsive formulations due to their sensitive hydrolysis with each unit of pH decrease [65,66]. Acetals are formed by condensation of aldehyde and alcohols. Under acidic conditions, acetal can be hydrolyzed into aldehyde and alcohols [67].

Zhong et al. conjugated acid biodegradable poly(acetal carbamate) (PAU) into a three block copolymer (PEG-PAU-PEG) and formulated to micelles by self-assembly for pH-triggered intracellular delivery of DOX. The three-block copolymer formed micelles had no cytotoxicity and performed good stability in the physiological pH. However, DOX could be triggered release in the acidic cytoplasm after uptake by endocytosis and induce apoptosis of cancer cells [68]. Pu and colleagues prepared tumor-pH-sensitive polymeric microspheres by using multi-block polyl-lactide (PLLA) with acetal bond as the backbone for efficient loading DOX. They reported better antitumor efficiency and prolonged life-span after intratumorally injection of the micelles compared to other control groups [69].

### 2.6. Orthoester

Orthoester is a functional group containing three alkoxy groups on the same carbon atom. Orthoester can be hydrolyzed into carboxylic acid and alcohol under acidic conditions [70].

Park et al. assembled micelles with copolymers bearing an acid-sensitive orthoester linkage, which composed of hydrophilic PEG and hydrophobic poly(γ-benzyl L-glutamate) (PBLG) and DOX. The DOX loaded polymeric micelles were stable under neutral conditions but could trigger the release of DOX for efficient therapy in the acidic tumor due to the acidic cleavage of orthoester [71].

## 3. pH-Sensitive Nanomaterials

In recent years, many administration routes have been widely developed for the medical application, and integrating the novel material engineering technologies with optimal administration route would enable their enhanced therapeutic efficiency and reduced side effects [72,73,74,75,76,77]. In terms of convenience, oral administration has great advantages. However, since drugs need to pass through the digestive tract during oral administration, and then arrive at the duodenum, jejunum, and ileum. The pH values in different parts of the gastrointestinal tract are different, that is, the stomach cavity is acidic, the pH range is 1–3, and the duodenum is alkaline [78]. When a drug enters the gastrointestinal tract by oral administration, the drug will be exposed to a series of acidic environments and biological enzymes, which may be degraded or inactivated. The tumor microenvironment also has features with acidic and other abnormal characteristics which may suppress the bioactivity of delivered drugs [79]. These required the novel drug carriers which can protect the drugs from degradation and metabolisms prior to their final acting places and triggered release of the therapeutic payloads in the abnormal pathological tissues according to the pH gradients or other conditions.

Nanomaterials for drug delivery have broad application prospects [41,80,81,82,83,84,85,86]. For example, the nanoparticles encapsulated with drug molecules can isolate drugs and environments that may degrade or inactivate drugs, providing a relative stable environment for drug molecules with expected bioavailability while passing through the complex in vivo environment. At the same time, since the nanoparticles allow them to penetrate the endothelium and capillary tubes, the use of nanoparticles to deliver drugs to target cells in inflammatory sites may be considered [87]. These carriers can release drugs in the extracellular environment, or within the cell through endocytosis. pH-responsive characteristics can be added into the nanoparticles which can be degradation or deformation under specific conditions, such as acidic conditions, thereby releasing the encapsulated drugs and acting on a specific location. After the changes of nanoparticles, the release of drugs from nanocarriers can be triggered based on the change of pH [88,89,90,91]. Compared with the ordinary nanoparticles, advantages of pH-responsive nanoparticles lie in applying acidic pH as a kind of external stimuli, resulting in the changes of drug release kinetics [92,93,94]. Herein, we present an overview of pH-sensitive nanomaterials [95] including hydrogel, liposomes, and polymer micelles.

### 3.1. Hydrogels

Hydrogels are composed of polymers that are crosslinked into a three-dimensional network [96,97,98]. Hydrogels can be made of synthetic or natural polymers and they are capable of imbibing large quantity of water or fluids [99,100]. There are a large number of hydrophilic groups on the polymer chain, such as -NH_2_, -OH, -COOH, and -SO_3_. With the increase of capillary action and osmotic pressure, hydrogels are relatively insoluble in the surrounding media due to the cross-linking between the polymer chains. The cross-linking in hydrogels is both physical (hydrogen bonding) and chemical (covalent bonding between functional groups) (Figure 2a). Hydrogel was first proposed in the 1960s by Wichterle and Lim using a hydrophilic network of poly(2-hydroxyethyl methacrylate) in contact lenses [101,102]. Hydrogels are valuable due to their easy fabrication process, small size, mouldability, immunity to electromagnetic radiation, and biocompatibility [100]. The physical characteristics of hydrogels can be tuned to provide specifications to match various applications in drug delivery systems or other aspects [103,104,105,106,107,108]. pH-responsive hydrogels been developed greatly over the past few years and shown to be very useful in biomedical applications, especially like targeted cancer treatment [109,110,111]. The use of pH-sensitive hydrogels can prolong drug release availability, and the synthesis of pH-sensitive hydrogels is also quick and cost-effective. In addition, pH-sensitive hydrogels have been investigated frequently for delivery of peptide drugs [112,113].

Currently, the use of pH-sensitive hydrogels for cancer therapy has been investigated as improving the efficiency of the released drugs [114,115,116,117,118]. Polymers commonly used to study pH-responsive hydrogels including polyacrylate, poly (*N*-isopropyl acrylamide), polyacrylate, *n*-vinyl caprolactam, sodium alginate, and carrageenan [119,120].

Some traditional hydrogels require the use of different substances, such as poly(N-isopropyl acrylamide) and poly(acrylic acid) attached to the PVF skeleton, fabricating hydrogels with dual temperature and pH response [119]. Alternatively, bivalent copper can be used in conjunction with alginate to form hydrogels by gel spheroidization. It is worth mentioning that the encapsulation system made by this method has been proved to maintain their structural integrity at pH 1.2, and the contents can only be released slowly at pH > 5, which suggests the stability of the encapsulation system under the conditions of simulated stomach, and release the contents slowly under the conditions of intestinal tract [123,124]. In addition to the traditional hydrogels, there are also novel hydrogels, such as gelation by crosslinking of poly (vinyl pyrrolidone) polymer and rotonic acid under γ-radiation [122]. The drug release rate from the gelation in acidic medium is much lower, while the release rate in neutral medium is faster. Therefore, similar to the former, it has the potential to act as a drug carrier to protect the release of drugs through the stomach in the intestinal tract. Some hydrogels also need to improve the stability of the pore wall to prevent swelling, such as the use of silica nanocomposites on top of polyn-isopropyl acrylamide [122]. Furthermore, the majority of the pH-responsive hydrogels [125] relies on the side chain by ionization, namely through the protonation of cationic groups or anionic groups, or the electric network with the surrounding environment, forming electrostatic repulsion [121] (Figure 2b,c).

In addition, dual temperature and pH-responsive hydrogels or dual magnetic and pH-responsive hydrogels were also expanding their medical application, allowing for more precise drug release behavior. Ren et al. synthesized the supramolecular hydrogels hybridized with magnetic nanoparticles (MNPs) and gold nanoparticles (AuNPs). These hybrid hydrogels which stimulated by temperature and pH showed reversible sol–gel transition [126] (Figure 3). There are also pH-responsive cellulose gels used as wound dressings based on hydrogels that could self-degrade on the mild acidic skin surfaces and have the ability to self-heal, which have broad application prospects. Amanda et al. showed that surface functionalization of cellulose nanocrystals (CNCs) with amine (CNC-NH_2_) moieties enabled the CNCs with pH-responsive characteristics. The transition to hydrogels as observed at higher pH degraded into dispersion aqueous at low pH. This could be incorporated into a poly(vinyl acetate) matrix to yield mechanically adaptive pH-responsive nanocomposite films [127] (Figure 4). 

### 3.2. Liposomes

In the 1960s, the first lipid-based vesicles for drug delivery was reported [128,129,130]. Liposomes [131] are a kind of biomimetic nanosome with a hollow structure consisting of phospholipid bilayers (Figure 5). The ability of liposomes to encapsulate both hydrophilic and hydrophobic drugs, coupled with their biocompatibility and biodegradability, make liposomes attractive vehicles in the field of drug delivery [132,133,134,135]. Liposomes are frequently used for delivery of drugs, antigens, vaccines [136], DNA, and/or diagnostic units [137,138,139,140,141]. The liposomes could be internalized by cells through different endocytic pathways and effectively deliver the encapsulated drugs into the cell matrix [128,129,142,143,144,145]. For their optimal in vivo application [146], the surface of liposomes can be functionalized with various hydrophilic polymers, such as PEG, to minimize reticuloendothelial system recognition and absorption [147,148]. The diameter of liposomes enables passive accumulation in tumor tissue for antitumor effect through the enhanced permeability and retention effect (EPR) [149,150,151,152]. The composition of liposomal systems can be easily modified to facilitate triggered release in response to environmental conditions. For example, pH-responsive liposomes are specifically designed to control the drug release in response to the acidic tumor microenvironment [153].

pH-responsive liposomes [154,155] represent a promising carrier for the loading of drugs for medical applications [153]. One common method to prepare pH-responsive liposomes is to use pH-sensitive components to fabricate liposomes. When these responsive units introduced into the liposomes, the liposome can be transformation or degradation in the acidic endosomes or environments of tumors, resulting in the trigger release of payload with pH-responsive behaviors [156]. In addition to the instability caused by pH-sensitive components, the stability of liposomes can also be affected by the integrated hydrophobic chains. These pH-liposomes have some significant advantages, such as low toxicity, simple preparation, and good biocompatibility as a result of the biocompatible degradable components [157,158].

pH-responsive liposomes [159] can function as the carrier of anti-tumor therapeutics for targeting delivery to the tumor and trigger release of the encapsulated drugs in the acidic environment. However, their use for in vivo application also presents great challenges. In general, the size of liposomes used for in vivo application is usually between 50 and 450 nm, and the liposomes within this diameter scale are easily removed by the RES system during blood circulation [157,160]. At the same time, although their structure and composition are similar to that of the cell membrane, liposomes are still easily recognizable as the antigens, and faced with the in vivo clearance by the immune system. To improve their in vivo fate for better therapeutic outcome, various surface modification strategies are developed such as introducing the PEG chains to the outside surface of liposomes, so as to prolong their in vivo circulation time [130,157,161,162] (Figure 6). In addition to the therapeutic purpose, pH-responsive liposomes are also becoming important tools for vaccine delivery. These liposomes are used to deliver small peptides or antibodies to produce an effective immune response and reduce their toxicity. Kim and colleagues successfully transferred the pH-sensitive liposomes loaded with cytotoxic T lymphocytes epitope peptides for tumor therapy, which resulted from the induced effective antigen-specific CTL responses after the responsive release of peptides in the acidic tumor [163].

### 3.3. Polymer Micelles

Polymer micelles are promising carriers for various kinds of drugs in medical application, thereby improving the efficacy of drugs and reducing side effects [164,165,166,167,168,169] (Figure 7). Micelles are usually formed by self-assembly of block polymers, which could be conjugated with different units such as polyethylene glycol and poly(amino acid) [169]. During the self-assembly of the polymers of micelles, therapeutic drugs such as DNA, RNA, proteins, and small molecular drugs with an abundant of functions could be encapsulated within the micelles [170,171,172,173,174]. 

Although polymer micelles have good stability in vivo, drugs released from the micelles usually in a very slow diffusion mode, which lead to a low concentration of free drugs in tumor cells and would suppress the therapeutic outcome. It is necessary to design polymer micelles to release drugs with desirable dynamic behavior in tumor sites. pH-responsive polymer micelles equipped with acidic trigger sections present an efficient drug delivery for cancer therapy [56], derived from the passive or active targeting potency and subsequent responsive release of payloads.

The design strategies of pH-sensitive polymer micelles mainly include:Protonated groups are introduced into the polymer chain through the mechanism of protonation/deprotonation [175,176,177,178,179]. The common functional groups of protonation include the amine group, imidazolyl group, sulfonic acid group, and carboxyl group. When such polymer micelles reach acidic target sites, rapid targeted release of the drug contained in the micelles would be triggered as the unstable micelle structures due to acidic precipitation/aggregation, or depolymerization.pH-sensitive chemical bonding arms are introduced between polymer chain segments or between polymers and drugs [180,181,182]. The acid sensitive connector arm refers to a molecule or group that can exist stably under neutral conditions but can hydrolyze quickly under weak acidic conditions, including imine bond [183], hydrazine bond, hydrazone bond [56,184,185,186], cisaconitamide, dimethyl maleamide [94,187], ether bond [56,188], orthate ester [70,188,189], polyacylaldehyde (ketone) [190], etc.

Pun et al. recently developed pH-sensitive polymer micelles fabricated with self-assembly of lytic peptide modified diblock copolymers (PPCs). They demonstrated that significantly enhanced in vitro and in vivo transfection efficiency with melittin-like peptide polymer conjugate encapsulated in the micelles [191], indicating further in vivo gene therapy mediated by the peptide encapsulated micelles. 

The surface modification of micelles, such as PEGylation, is important for their function [56]. The long chain of PEG may affect the uptake of micelles by cells for subsequent biological effects. Other modification strategies could be considered for improved uptake by cells, as a tactic folate acid and its variants can be used to modify the surface of micelles for improving cellular uptake, as folic acid receptors are usually highly expressed on many types of cancer cells. Xiong et al. successfully designed folate-conjugated crosslinked biodegradable micelles with poly(ethylene glycol)-b-poly(acryloyl carbonate)-b-poly(D,L-lactide) (PEG-PAC-PLA) and folate-PEG-PLA (FA-PEG-PLA) block copolymers for receptor-mediated delivery of paclitaxel (PTX) into KB cells for cancer therapy [192](Figure 8).

## 4. Conclusions and Future Prospects

Over the last few decades, engineering biomaterials have strived to improve the therapeutic outcome with reduced effects, and great progress has been made in the application in drug delivery systems for medical applications, especially for cancer therapy. Several biomaterials with pH sensitivity have shown great potential for constructing effective drug delivery systems. In this review, we present an overview for pH-sensitive bonds and pH-sensitive nanomaterials for drug delivery in the past two decades. pH-sensitive biomaterials for drug delivery are a research hotspot in the fields of biochemistry and medicine, and great progress has also been made in recent years. At the same time, most of the research is still in the experimental stage, and there are still many problems that need to be solved for translational application, such as the biocompatibility of some pH-sensitive biomaterials, mass production with controllable quality, and potential toxicology of the carriers. Using pH-sensitive chemical bonds to design and prepare pH-sensitive biomedical materials based on acidic microenvironments of diseases would greatly improve the biocompatibilities and reduce potential toxicity of biomaterials. Microfluidics to screen the uniform size of biomaterials would greatly improve the consistency of synthesized samples with different batches. 

## Figures and Tables

**Figure 1 molecules-25-05649-f001:**
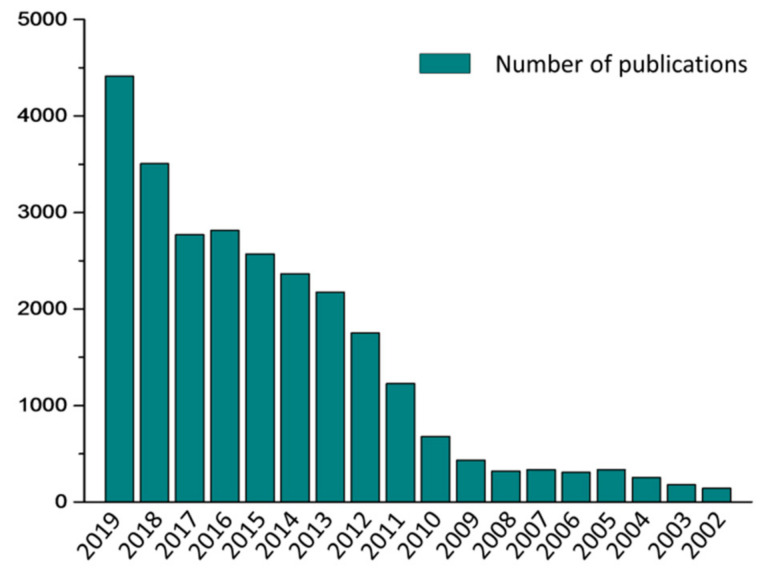
Statistics on the number of published journals on pH-sensitive biomaterials over the past 20 years.

**Figure 2 molecules-25-05649-f002:**
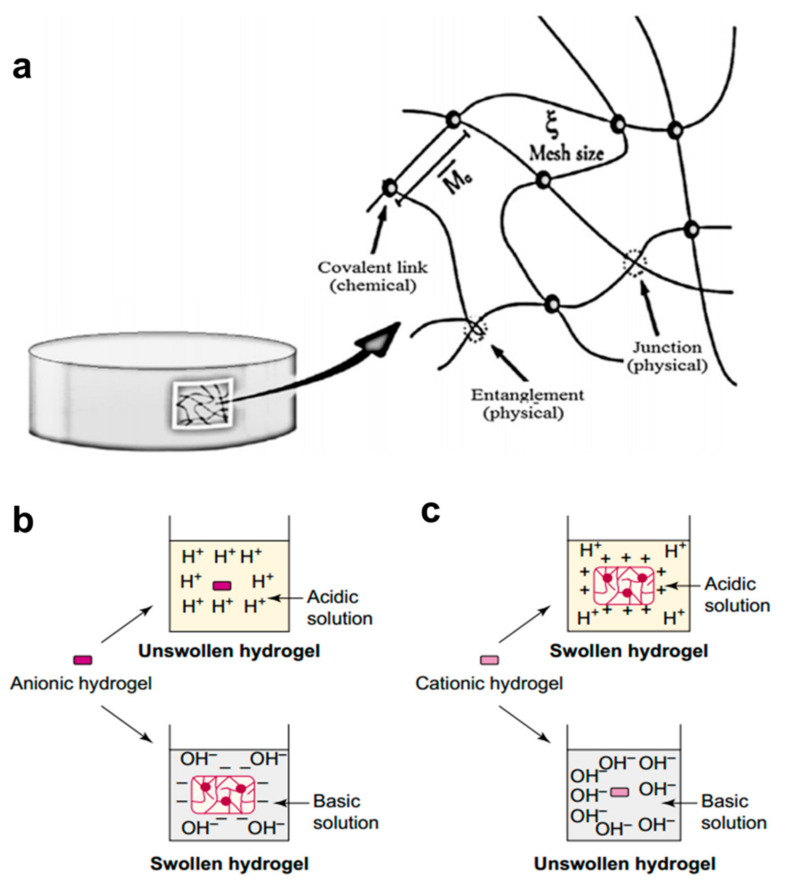
Hydrogels: (**a**) structural chemistry of a hydrogel [121]; (**b**,**c**) the pH-responsive swelling of anionic (**b**) and cationic (**c**) hydrogels [122].

**Figure 3 molecules-25-05649-f003:**
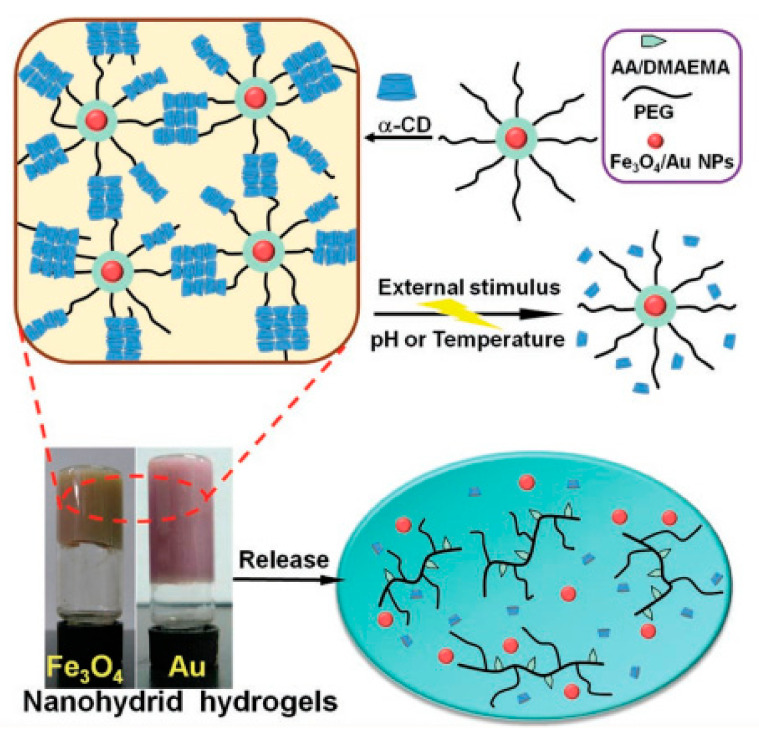
Synthesis of the supramolecular hydrogels hybridized with inorganic MNPs and AuNPs and its release [126].

**Figure 4 molecules-25-05649-f004:**
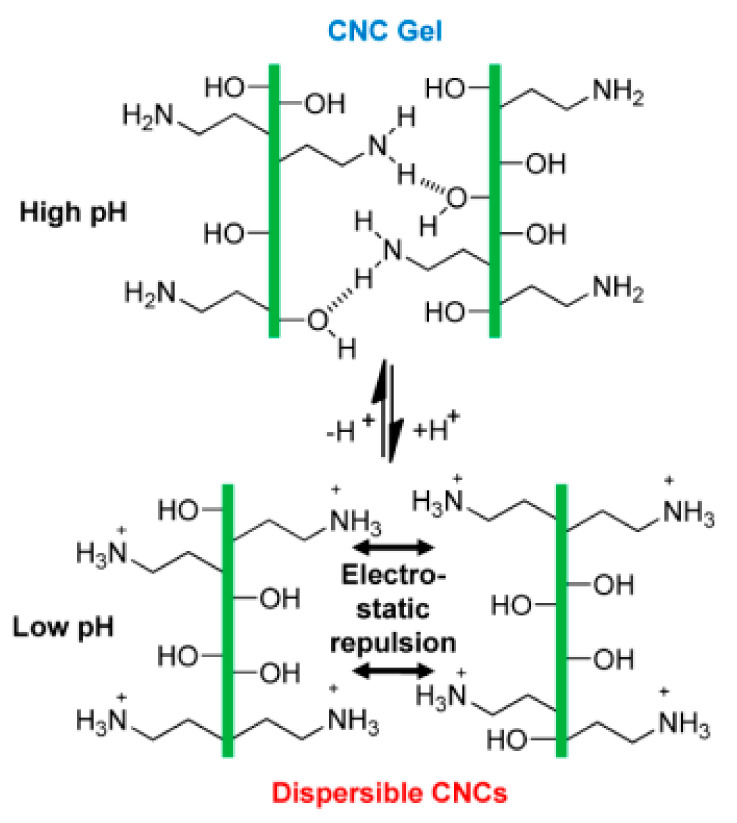
Schematic representation of the proposed interactions of CNC−NH_2_ at high and low pH [127].

**Figure 5 molecules-25-05649-f005:**
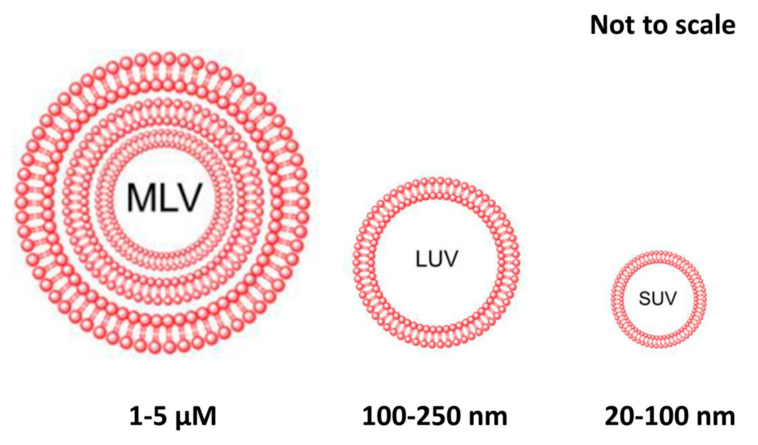
Classification of liposomes based on the lamellarity: Multilamellar Vesicles (MLV) with many lipid bilayers ranges from 1–5 µm in size. Large Unilamellar Vesicle (LUV) is about 100–250 nm with single lipid bilayer. Small Unilamellar Vesicles (SUV) with size range 20–100 nm consists of a single phospholipid bilayer (reused with modification) [157].

**Figure 6 molecules-25-05649-f006:**
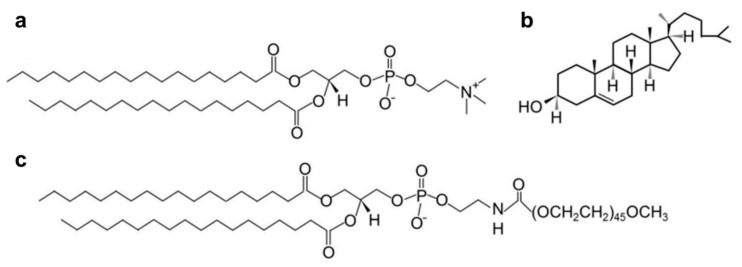
Structures of common liposomal constituents: (**a**) 1,2-distearoyl-sn-glycerophosphocholine (DSPC); (**b**) cholesterol; (**c**) 1,2-distearoyl-sn-glycero-3-phosphoethanolamine polyethylene glycol (DSPEPEG) [157].

**Figure 7 molecules-25-05649-f007:**
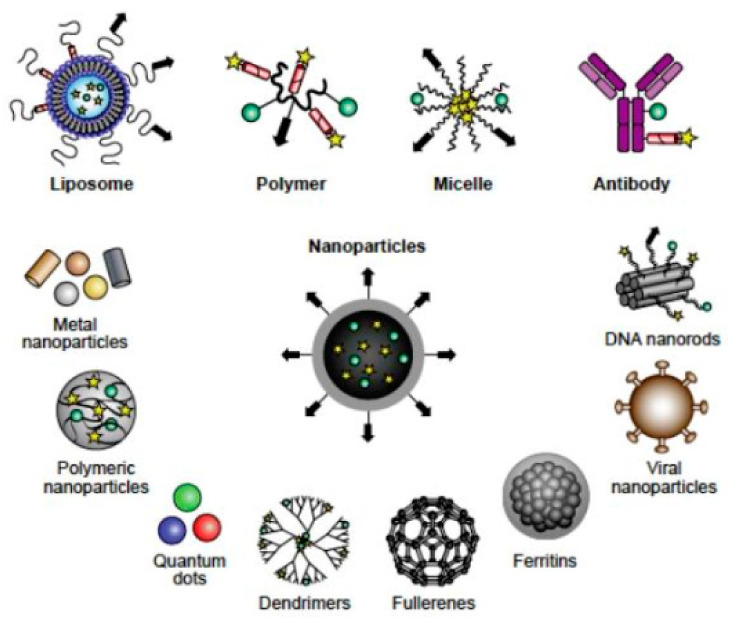
Biocompatible nanocarriers include those composed of liposomes, polymers, micelles, antibodies, nanoparticles composed of metals, and other biological molecules or combinations [167].

**Figure 8 molecules-25-05649-f008:**
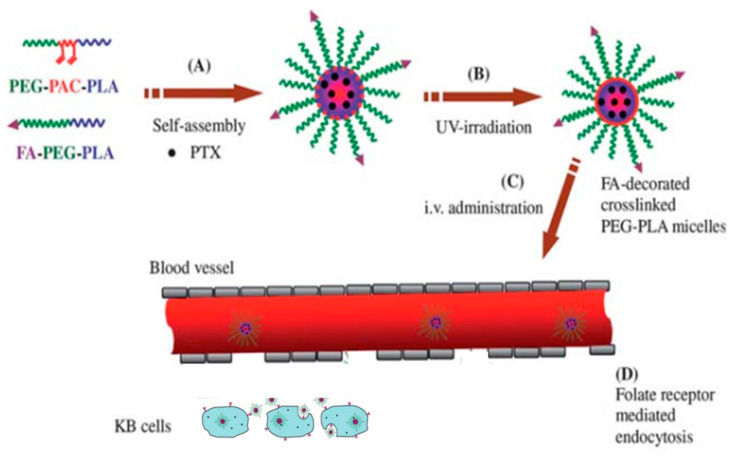
Illustration of folate-conjugated interfacially crosslinked biodegradable micelles for targeted delivery of paclitaxel [192].

**Table 1 molecules-25-05649-t001:** pH-sensitive chemical bonds and release mechanisms in the acidic conditions.

pH-Sensitive Bonds	Chemical Mechanisms	Applications in Ref.
Imine	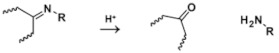	[34,35,36,37,38,39,40,41,42]
Hydrazone	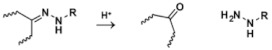	[43,44,45,46,47,48,49,50,51]
Oxime	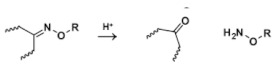	[52,53,54,55,56]
Amide	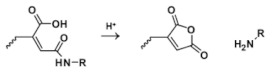	[57,58,59,60,61,62,63,64,65]
Acetals	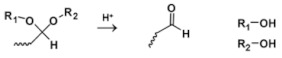	[66,67,68,69,70]
Orthoester	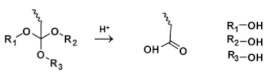	[71,72]

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
