# Peer review of "pH-Sensitive Biomaterials for Drug Delivery"

_molecules, 2020, doi:10.3390/molecules25235649_

Round 1
Reviewer 1 Report
Comments
This is a mini-topical review on pH-sensitive biomaterials for drug delivery. Studies of pH biomaterials on biomedical application and therapy are timely and topical. This work is quite comprehensive and I only have some minor comments for further improvement:
- In the Abstract and Introduction section, the authors should state what is the aim to conduct such review on pH biomaterial? What is the purpose of the review?
- Can the authors give some statistics on the number of publications regarding pH biomaterials in recent years?
- Table 1 starts the reference from [20] while the text above is [12]. The reference number between the text and table does not match.
- L139: For application of nanomaterials on therapy, some updated important references such as Siddique et al (Nanomaterials, 2020;10:1700) should be included.
- L151: For application of nanomaterials on drug delivery, some updated important references such as Siddique et al (App Sci, 2020;10(11): 3824) should be included.
- The Conclusion section should include some statements regarding future works based on the results of the review.
- Figure 4: the caption should mention the drawing is “not to scale”.
Author Response
Reviewer 1 : This is a mini-topical review on pH-sensitive biomaterials for drug delivery. Studies of pH biomaterials on biomedical application and therapy are timely and topical. This work is quite comprehensive and I only have some minor comments for further improvement:
- In the Abstract and Introduction section, the authors should state what is the aim to conduct such review on pH biomaterial? What is the purpose of the review?
Reply: Many thanks for the good question and suggestion. We have already added the content into the corresponding part.
Abstract section:
…As many diseased tissues are feathered with acidic characteristics microenvironment,…
Introduction section:
…“pH-sensitive biomaterials has been developed rapidly in the past decades and great achievements have been made. As shown in Figure 1, we have summarized the number of publications about pH-sensitive biomaterials in the past 20 years(Figure 1). In this review, we focuses on the pH-responsive linkage bonds and pH-sensitive nanomaterials for medical applications, especially for cancer therapy. Firstly, we summarize some common pH-sensitive biomaterials based on different types of pH-sensitive bonds. In the following, we summarize pH-sensitive nanomaterials for drug delivery purpose.”…
- Can the authors give some statistics on the number of publications regarding pH biomaterials in recent years?
Reply: Many thanks for the good question and suggestion. We summarize the publication regarding pH-responsive biomaterials from the Web of Science, as describle in Figure 1.
Figure 1. The Statistics on the number of published journals on pH-sensitive biomaterials over the past 20 years.
- Table 1 starts the reference from [20] while the text above is [12]. The reference number between the text and table does not match.
Reply: Many thanks for the good attention. It was our mistake. We have already revised it as in Table 1:
- L139: For application of nanomaterials on therapy, some updated important references such as Siddique et al (Nanomaterials, 2020;10:1700) should be included.
Reply: Many thanks for the good suggestion. As suggested, we added the Siddique et al (Nanomaterials, 2020;10:1700) and other related references in the pH-responsive biomaterials section.
- L151: For application of nanomaterials on drug delivery, some updated important references such as Siddique et al (App Sci, 2020;10(11): 3824) should be included.
Reply: Many thanks for the good suggestion. As suggested, we added the Siddique et al (App Sci, 2020;10(11): 3824) labeled reference[87] in the corresponding position(L168). And we also added some relative references in recent years in this review.
- The Conclusion section should include some statements regarding future works based on the results of the review.
Reply: As suggested, we have already added the content into the corresponding part.
Using pH-sensitive chemical bonds to design and prepare pH-sensitive biomedical materials based on acidic microenvironment of diseases would greatly improve the biocompatibilities and reduce potential toxicity of biomaterials. Microfluidics to screen the uniform size of biomaterials would greatly improve the consistency of synthesized samples with different batches.
- Figure 4: the caption should mention the drawing is “not to scale”.
Reply: Many thanks for the good suggestion. As suggested, we have already added it in Figure 4.

Reviewer 2 Report
In this manuscript are presented pH-Sensitive Biomaterials and their applications. For a review it must be more comprehensive and present the latest data in the field. Please add the following:
- details each chapter in more detail: for exemple in hydrogels describe also the collagen, hyaluronic acid, gellatin etc.
- In figure 6 you put a lot of types of nanocarriers, please describe them also in text
- fig. 7 the cells are larger then blood vessel, please modified the figure by respectin the biological proportions
- editing of English language
- add more references, a minimum of 200 is required for a good review
Author Response
Reviewer 2: In this manuscript are presented pH-Sensitive Biomaterials and their applications. For a review it must be more comprehensive and present the latest data in the field. Please add the following:
- Details each chapter in more detail: for exemple in hydrogels describe also the collagen, hyaluronic acid, gellatin etc.
Reply: Many thanks for the good suggestion. As suggested, we have supplemented the content of each part appropriately and added relevant references.
- In figure 6 you put a lot of types of nanocarriers, please describe them also in text
Reply: Many thanks for the good suggestion. As suggested, we have supplemented the related content and relevant references for each carrier for revised figure 6 (relabeled Figure 7).
- 7 the cells are larger than blood vessel, please modified the figure by respectin the biological proportions
Reply: As suggested, we have revised the figure according to your valuable suggestions.
- Add more references, a minimum of 200 is required for a good review.
Reply: Many thanks for the good suggestion. As suggested, we have added contents and relevant references as highlight in yellow background.

Round 2
Reviewer 1 Report
I am satisfied with the corrections and additional contents made by the authors in the revision. The presentation and quality of the paper have been improved.
Reviewer 2 Report
The manuscript has been improved from the previous form and can be considered for publication